# Monitoring the impact of confinement on hyphal penetration and fungal behavior

Yi-Syuan Guo[1,2,3], Julian A. Liber[4], Jennifer L. Morrell-Falvey[2], Gregory Bonito[4], Scott T. Retterer[2,3]*

**1** Environmental Molecular Sciences Division, Pacific Northwest National Laboratory, Richland, Washington, United States of America, **2** Biosciences Division, Oak Ridge National Laboratory, Oak Ridge, Tennessee, United States of America, **3** Center for Nanophase Materials Sciences, Oak Ridge National Laboratory, Oak Ridge, Tennessee, United States of America, **4** Department of Plant Soil and Microbial Sciences, Michigan State University, East Lansing, Michigan, United States of America

* rettererst@ornl.gov

**Data Availability Statement:** All relevant data are within the manuscript.

**Funding:** This work was supported by the Office of Science, Biological and Environmental Research, as part of the Plant Microbe Interfaces Scientific

## Abstract

Through their expansive mycelium network, soil fungi alter the physical arrangement and chemical composition of their local environment. This can significantly impact bacterial distribution and nutrient transport and can play a dramatic role in shaping the rhizosphere around a developing plant. However, direct observation and quantitation of such behaviors is extremely difficult due to the opacity and complex porosity of the soil microenvironment. In this study, we demonstrate the development and use of an engineered microhabitat to visualize fungal growth in response to varied levels of confinement. Microfluidics were fabricated using photolithography and conventional soft lithography, assembled onto glass slides, and prepared to accommodate fungal cultures. Selected fungal strains across three phyla (Ascomycota: *Morchella sextalata*, *Fusarium falciforme*; Mucoromycota: *Linnemannia elongata*, *Podila minutissima*, *Benniella*; Basidiomycota: *Laccaria bicolor*, and *Serendipita* sp.) were cultured within microhabitats and imaged using time-lapse microscopy to visualize development at the mycelial level. Fungal hyphae of each strain were imaged as they penetrated through microchannels with well-defined pore dimensions. The hyphal penetration rates through the microchannels were quantified via image analysis. Other behaviors, including differences in the degree of branching, peer movement, and tip strength were also recorded for each strain. Our results provide a repeatable and easy-to-use approach for culturing fungi within a microfluidics platform and for visualizing the impact of confinement on hyphal growth and other fungal behaviors pertinent to their remodeling of the underground environment.

## Introduction

The synergies between plant hosts and their evolving microbiome intricately shape local microenvironments within the rhizosphere. These spatial and temporal developments within the micropores of soil drive nutrient cycling, modulate soil hydrodynamics, dictate microbial activities, and potentially supply benefits to the plant host such as enhanced growth and protection from pathogens [1]. The inherent heterogeneity across this collection of

Focus Area (https://pmi.ornl.gov), Department of Energy, BER, ERKP730. Fabrication of Microfluidic Devices was performed at the Center for Nanophase Materials Sciences, A DOE Nanoscience User Facility, Department of Energy, BES ERKCZ01.

**Competing interests:** The authors have declared that no competing interests exist.

microenvironments and the lack of accessibility for sampling and visualization make the rhizosphere a daunting system to study. Uncovering specific relationships among plants, microbes, and the soil environment requires innovative approaches [2–4]. In particular, understanding the assembly and organization of microbiomes during plant growth and development and identifying the molecular signal exchanges that shape these interactions requires a systematic method for quantifying the growth and response of soil microbiomes to such drivers.

Fungi have a tremendous impact on the physical and chemical makeup of their local environment and are important members of soil and plant microbiomes. Expansion from the tips of their filamentous bodies enables fungi to grow and extend thread-like tubes, known as hyphae, through confined and complex spatial networks. Within the soil, this allows fungi to rapidly explore and scavenge for resources, directly and indirectly facilitating the transport of nutrients and water underground [5]. Moreover, recent studies have highlighted the role that fungal hyphae play in bacterial transport whereby fungal hyphae function as 'super-highways' that influence and accelerate the distribution of bacteria along established 'routes' allowing bacteria to rapidly reach new areas of the soil that would not otherwise be accessible [6]. This alters local microbial consortia/communities and can consequently change the development of growing plant hosts in the vicinity [7].

Beyond navigating, traversing, and filling the complex open spaces within the soil, fungi can actively remodel their local soil environment. Serving as decomposers, fungi produce extracellular enzymes that break down complex organic compounds in the soil into smaller molecules, participating in the cycling of carbon and nutrients and further modifying local soil chemistry and porosity [8]. Fungi are also able to physically bind soil particles together. This alters water infiltration and soil-water holding capacity at the pore scale [9]. Despite their impact on the underground network in which they thrive, methods for visualizing and phenotyping soil fungi in their native environment are limited.

Microfluidics has enabled the creation of a series of well-defined microarchitectures that can be replicated and applied to examine how differences in local structure impact hyphal growth, specifically examining the role that confinement and connectivity have on the ability of fungal species to navigate complex soil-like environments. The well-defined fluid space, small volumes, and consequent laminar flows in microfluidics allow for rapid and predictable control of the local chemical environment, opening the door to intentionally creating spatial and temporal concentration gradients of specific nutrients or signals to which fungi can respond. Notably, the glass and silicone used to construct most microfluidic platforms are transparent to visible light, gas permeable (allowing for controlled oxygenation), and often confine growth to a single in-plane focal volume making them ideal for time-lapse, brightfield, and fluorescence microscopies.

Microfluidics has provided a window for scientists to systematically study and view fungal behaviors in real time [10–12]. By tailoring well-defined geometries and using high-resolution microscopy, scientists have used microfluidics to visualize processes that include the response of fungal hyphae to physical architectures, spore germination, and other microbial-driven processes in well-defined geometry [13, 14]. For example, Held et al. (2019) identified that constraining geometries in microenvironments influence the spatial partitioning of the Spitzenkörper-microtubule system in *Neurospora crassa* elucidating the role that intracellular processes can play in directing fungal growth around obstacles [15]. Likewise, Aleklett et al. (2019) showed that soil fungi interact and explore habitat geometry by turning at various angles, navigating around circles, aggregating at corners, or branching under solvent-free incubation [16]. Such behaviors have a potential impact on cross-kingdom activity, with fungi promoting bacterial biomass accumulation in the microstructures [17]. In another study, different fungal strains exhibited a trade-off between cell polarity and growth rates when passing

through narrow channels, suggesting that spatial confinement may affect fungal species differently [18]. Similarly, Baranger et al., examined the impact of confinement on hyphal extension, noting significant decreases in extension rates under confined conditions. However, their work also noted significant differences between the two species examined with respect to their response to nutrient limitations [19].

In the work presented here, engineered microhabitats with well-defined microfluidic networks in poly-dimethylsiloxane (PDMS) were used to control local confinement and enable real-time quantitative phenotyping of multiple fungal species. Methods for controllably introducing, culturing, and capturing hyphal development using time-lapse microscopy and image analysis were developed and are described herein. This study demonstrates that the engineered microhabitat and methods are applicable to a broad range of soil fungi across multiple phyla despite significant differences in growth capacity and unique behaviors across the collection. Fungal strains used in this study were chosen to represent a range of phylogenetic diversity and observable differences in hyphal dimensions, branching frequency, spore formation, and growth rates. The precise control of habitat structure combined with direct visualization via time-lapse microscopy allowed quantitative analysis that revealed subtle changes in the impact of confinement on hyphal extension while also allowing observation of unique responses from different species.

## Methods

### Fungal species and growth media

Seven species of soil fungi were selected across three phyla for use in this study (Table 1). Isolates were provided by Bonito Lab. Each isolate was individually inoculated on potato dextrose agar (PDA, BD Difco™) plates (pH = 5.6 ± 0.2) and grown for 5–10 days at 25°C. Potato dextrose broth (PDB, pH = 5.1± 0.2, BD Difco™) was prepared according to the manufacturer's instructions for use within microfluidics.

**Table 1. Summary of traits and fungal behaviors of each strain.**

| | | | | Hyphal width (µm) | Incubation time (h) | Travel time (h) | Penetration Rates (µm min$^{-1}$) | | | Macro-scale | Key behaviors |
|---|---|---|---|---|---|---|---|---|---|---|---|
| | | | | | | | 5-µm width | 10-µm width | T-test value (p) | | |
| 1 | Ascomycota | *Morchella sextalata* | BR417Y | 7 ± 1 | 17–20 | 11–20 | 1.7 ± 1.3 | 2.0 ± 1.7 | 0.01 | 18 ± 3.9 | • Lateral branching<br>• Tip force strength |
| 2 | | *Fusarium falciforme* | MC67 | 3 ± 1 | 6–33 | 7–13 | 2.1 ± 0.7 | 1.9 ± 0.8 | < 0.005 (0.0045) | 6 ± 1.1 | • Linear penetration<br>• Tip force strength<br>• Strong mechanical property of spore |
| 3 | Mucoromycota | *Linnemannia elongata* | NVP64- | 3 ± 1 | 8–28 | 9–24 | 0.9 ± 0.9 | 1.8 ± 1.0 | < 0.005 (5.92E-31) | 10 ± 1.2 | • Peer movements |
| 4 | | *Podila minutissima* | AD051- | 3 ± 1 | 5–40 | 9–22 | 0.9 ± 0.8 | 1.6 ± 1.3 | < 0.005 (3.9E-13) | 7 ± 2.2 | • Peer movements |
| 5 | | *Benniella* | JL62 | 3 ± 1 | 5–24 | 6–17 | 2.4 ± 2.3 | 1.9 ± 1.4 | < 0.005 (0.0009) | 20 ± 2.1 | • Fastest grower<br>• Peer movements |
| 6 | Basidiomycota | *Laccaria bicolor* | S238N | 3 ± 1 | 21–86 | 90–240 | 0.08 ± 0.08 | 0.05 ± 0.05 | < 0.005 (7E-06) | 1 ± 0.3 | • High elasticity<br>• Tip force strength |
| 7 | | *Serendipita* sp. | NB90 | 2 ± 1 | 23–40 | 50–70 | 0.5 ± 0.4 | 0.2 ± 0.2 | < 0.005 (2E-186) | 5 ± 0.6 | • Pillar hugging<br>• Spore producing |

## Macroscale elongation measurements

Individual fungal plates were incubated at 23–24˚C. Each strain was streaked individually onto two plates. This process was repeated 4 times, resulting in a total of eight plates per species. A coordinate grid was drawn on the back of each Petri dish using a fine-tipped marker. The grid consisted of horizontal and vertical lines intersecting at regular intervals (major: 1 cm; minor: 0.5 cm), creating a reference system. To initiate the experiment, the fungal inoculum was punched as 3 mm, transferred, and placed at the center of each petri dish filled with PDA. The horizontal and vertical radial distances were measured from the point of inoculation to the edges of the colony every 24 hours. The average, maximum, and minimum growth rates were calculated as linear growth rates in μm per minute. The fungi were allowed to grow until reaching the edges of the Petri dishes or until they stopped growing.

## Microfluidics design and microfabrication

The microfluidic network consists of two inoculation ports on opposite sides of a buffering chamber. The inoculation ports are separated from the central chamber by 15 linear, parallel channels. Inoculation ports are 2-mm diameter with additional circular pillars spaced at regular intervals across open areas to keep the larger open areas from collapsing. The linear channels are 1000-μm in length and 6-μm in height (Fig 1). Devices were created with linear channels that were either 5μm or 10 μm wide. 100-μm markers were spaced at regular intervals along the linear channels to allow rapid measurement of penetration through the channels using digital image analysis.

Microfluidic masters were fabricated using conventional photolithography methods. A soda-lime glass/chrome mask was fabricated with the desired design. Clean, single crystal, 4-inch diameter silicon wafers were spin-coated with MicroPrime P20 adhesion promoter (Shin-Etsu MicroSi) to promote photoresist adhesion. NFR (JSR Micro, Inc., Sunnyvale, CA, United States), a negative photoresist, was spin-coated onto the wafers at 3000 RPM, soft-baked on a hot plate (90 ˚C, 90 s), exposed to 365-nm light at a dose of 50 mJ per cm^2 and then post-exposure baked on a hot plate at 115 ˚C for 90 s. Resist was developed in Microposit® MF® CD-26 developer (Shipley Company, Marlborough, MA, United States) for

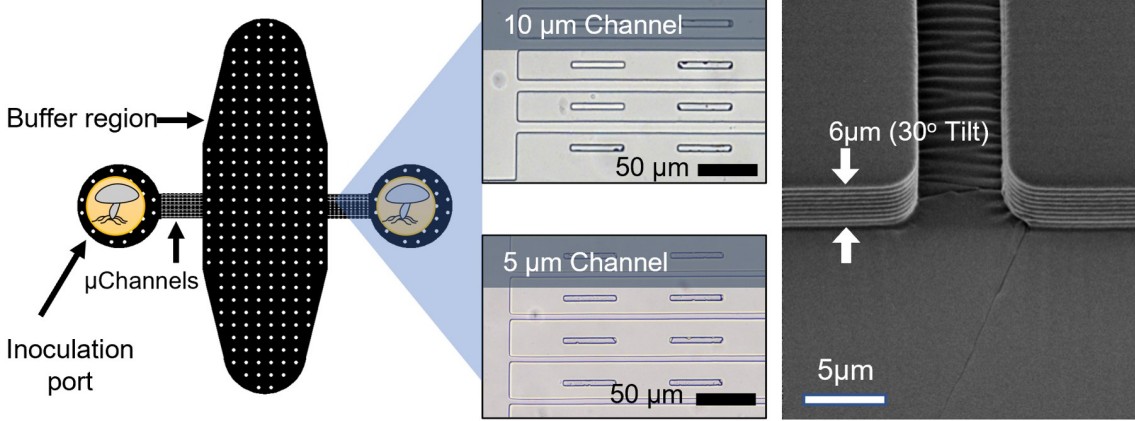

**Fig 1. Microfluidics design and its geometry.** (Left) Microhabitats consist of a central buffer region and two inoculation ports connected by a series of 15 microchannels. (Center) Optical micrographs show the entry region of the microchannels that connect the inoculation ports to the buffer channel. 5 and 10 μm channels were prepared with 50μm markers along their length. (Right) Channels were etched to a depth of 6 μm as shown in the scanning electron micrograph.

approximately 1 minute then rinsed with deionized water and blown dry with nitrogen. The wafers were etched for 8–9 cycles using a modified Bosch process (3s polymer deposition, 10 s etch) to produce the microfluidic channel with a profile height of 5.7–6.2 μm. Resist was removed using an oxygen plasma cleaning process. These 'masters' were treated with air plasma (2 min) in a Harrick Plasma Cleaner prior to vapor silanization with trichloro (1H,1H,2H,2H-perfluoro-n-octyl) silane. For vapor silanization, masters were placed in a glass petri dish with 20 μL of trichloro (1H,1H,2H,2H-perfluoro-n-octyl) silane. The glass petri dish with the master and droplet of silane reagent was left on a hotplate at 90∘C for 120 min. This treatment prevents polydimethylsiloxane (PDMS) adhesion to the microfluidic master during the PDMS molding process.

Microfluidic devices were fabricated using standard soft lithography methods. Briefly, poly-dimethylsiloxane (PDMS, Sylgard 184, Dow Corning, Midland, MI), a two-part silicone elasto-mer, and curing agent were mixed (5:1 ratio, respectively) and poured over the master, then were degassed in a vacuum chamber until no bubbles were visible. Molds were then cured at 70∘C for 3 h. PDMS replicas were cut out, removed from the master, and trimmed using a scal-pel. A 1.5-mm biopsy punch (Integra® Miltex) was aligned by eye and used to punch a hole in each inoculation port. PDMS castings were cleaned with Scotch tape. Once clean, the PDMS devices were treated with air plasma for 1 min, bonded onto plasma-cleaned glass slides, and placed into the oven to complete bonding (70˚C, 5 min).

After curing and bonding, the PDMS microdevices on glass slides were sterilized by autoclaving (121 ˚C, 20 min) or, alternatively, placed in a fume hood and UV sterilized for 30 minutes after loading with culture media.

## Microfluidic use and incubation

10 μL of culture media was added to each device using pipette tips. Filled devices were then sterilized under UV irradiation in a biosafety cabinet for 30 minutes. Media movement through devices was confirmed visually. Later, a 1 mm diameter biopsy punch with a plunger (Integra®) was used to remove 'fungal plugs' from agar-filled Petri dishes with fungi growing on the surface of the agar. These fungal plugs were placed into the inoculation ports of sterile devices with the fungi-covered surface of the agar plug against the glass slide as shown in Fig 2. A 1.2-mm diameter PDMS plug was then placed into the port to seal the device. The inocu-lated microfluidics were placed and fixed into a 100-mm petri dish using scotch tape (3M®) surrounded by DI water or dampened tissue. Dishes were covered with a lid and sealed closed with parafilm to prevent contamination and dehydration during incubation.

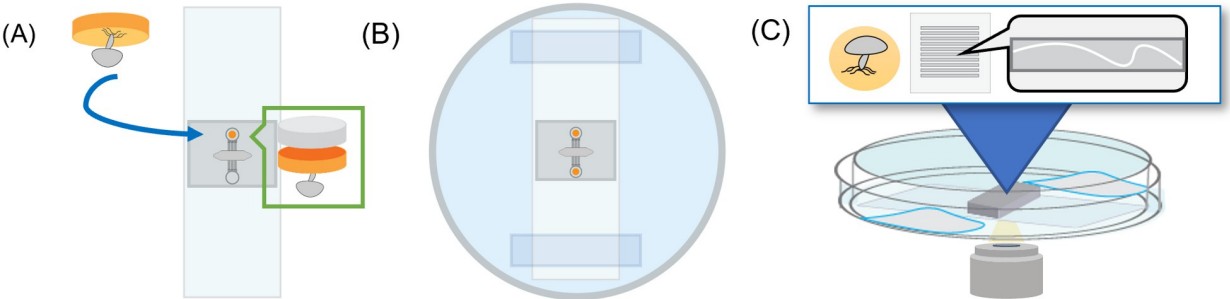

**Fig 2.** Fungal microfluidics setup (A) Fluidic devices bonded to a glass slide are inoculated with a plug of fungal agar sealed with a small PDMS plug. (B) The devices are then fixed to the bottom of a petri dish with scotch tape. Water or damp tissues are added around the devices, a cover is added, and the device is sealed with parafilm. (C) Devices are placed into a 3D-printed dish holder and imaged at set intervals in an inverted microscope for the duration of the experiment.

## Time-lapse microscopy and image acquisition

The time-lapse images of fungal microfluidics were collected at 10x, or 20x magnification using an inverted microscope (Olympus IX-51 inverted microscope, Nikon Ti-U inverted microscope, and Nikon swept field confocal microscope). Growth was monitored following inoculation and was continued until fungi exited the linear fluidic channels or stopped growing. Each of the species in Table 1 was tested in both the 5μm and 10μm wide linear channels in triplicate to quintuplicate. A detailed summary of the numbers of hyphae/microchannels analyzed and experimental replicates performed is provided in Table 2.

Three to five experimental replicates (n, Table 2) were performed for each channel width and fungal species. For each experimental replicate, the number of hyphae/channels analyzed is shown in Table 2 (Events Analyzed) for sets of experiments performed in the 5-μm and 10μm-channel devices. For example, for the 5μm wide channel experiments conducted with *Morchella sextelata*, three experimental replicates were performed. 11 channels were analyzed from the first replicate, 7 were analyzed in the second, and 12 in the third.

Once a leading hypha entered a linear channel, its penetration into the channel was measured periodically until it exited the channel or stopped growing (*L. bicolor*: every 24h; *Serendipita* sp.: every 2h; others: every 1h). Each channel occupied by hyphae was measured between the channel entrance and the tip of the leading hyphae using the built-in measurement tool from Nikon NIS Elements and the 50-μm markers printed below each channel as a reference. Fungal penetration rates were calculated from each image set in units of μm per minute.

All the measured values of each set were collected, grouped, and are shown as histograms of instantaneous penetration rates in Fig 4. The total number of measurements made for each data set are shown in Table 2 (Total # Measured). The average value is shown as mean ± standard deviation in Table 1. An unpaired Student's t-test was performed to compare the results obtained between the data collected in 5-μm and 10-μm-wide channels for each species. P-values less than 0.005 were statistically significant.

## Results

### Characterization of fungal growth in engineered microhabitats

Fungal isolates representing 3 major fungal phyla were inoculated individually. Each species was cultured using potato dextrose agar steadily over 5 and then a 1-mm of fungal biomass from the growing edge was transferred into an inoculation port of microfluidics. The fungal behaviors were observed over extended periods, ranging from a few hours to multiple days. The morphology of the fungal hyphae and their growing mycelial networks were monitored in Petri dishes containing potato dextrose agar, within 'open' microfluidic geometries prior to

**Table 2. Detailed summary of the analyzed data in this study.**

| Phylum | Species | | 5 μm | | | 10 μm | | |
|---|---|---|---|---|---|---|---|---|
| | | | n | Events Analyzed | Total # Measured | n | Events Analyzed | Total # Measured |
| Ascomycota | *Morchella sextelata* | BR417Y | 3 | 11,7,12 | 227 | 4 | 10,9,31,9 | 389 |
| | *Fusarium falciforme* | MC67 | 3 | 14,15,13 | 266 | 4 | 12,15,5,15 | 322 |
| Mucoromycota | *Linnemannia elongata* | NVP64- | 3 | 13,10,13 | 341 | 4 | 13,10,15,8 | 314 |
| | *Podila minutissima* | AD051- | 3 | 15,7,14 | 315 | 3 | 14,6,14 | 229 |
| | *Benniella* | JL62 | 3 | 14,9,29 | 239 | 4 | 15,15,20,14 | 398 |
| Basidiomycota | *Laccaria bicolor* | S238N | 5 | 15,29,14,15,11 | 329 | 4 | 14,13,14,14 | 167 |
| | *Serendipita* sp. | NB90 | 3 | 30,26,25 | 2116 | 3 | 6,19,15 | 2232 |

entering microhabitat channels (Fig 3A) and during penetration, under confinement, through the microchannels. Species exhibited differences in the density of their mycelial network, hyphal elasticity, hyphal width, branching pattern (linear or lateral branching), and tip force generation (as noted by breaking of the PDMS bonding). Fungal mycelia were punched, transferred, and inoculated using microfluidics habitats and observed using microscopy. Representative microscopic images of events are shown in Fig 3B.

In this study, *M. sextalata* demonstrated significant amounts of lateral branching when growing in the engineered microhabitat (Fig 1B). In contrast, we observed that *F. falciforme* branched very little as it moved through the channels (Fig 1B). Somewhat surprisingly, both *M. sextalata* and *F. falciforme* were able to generate enough force to disrupt the bonding between the molded PDMS and glass substrate. *L. bicolor*, the slowest-growing fungal isolate within the tested species, showed elasticity and flexibility with highly curved fungal hyphae that seemed to have sparse anchoring to the substrates. *S.* sp. had characteristic wavy and narrow hyphae, and exhibited denser growth near physical obstacles, 'hugging' obstacles such as pillars. Isolates from the phylum Mucoromycota, which includes *L. elongata*, *P. minutissima*, and *Benniella*, demonstrated peer movement, in which we observed that multiple hyphae branch and explore neighboring regions, simultaneously, in a seemingly synchronized manner.

## Measurement of penetration rates through microchannels

The average hyphal width of all the species that were analyzed was $3 \pm 1$ μm, less than the width and height of the channels in the microhabitat and is reported in Table 1. A notable exception to this is *M. sextalata*, which was measured to have an average hyphal width of $7 \pm 1$ μm. Importantly, differences in the amount of hyphal branching within the microchannels (not quantified), as well as peer movement where multiple hyphae enter the same channel can result in additional 'packing' and increases in confinement.

The position of the tips of leading hypha traveling through the 5 and 10 μm wide channels was measured at set intervals for each of the species listed in Table 1. The penetration length was calculated as the distance of the lead hypha tip from the entrance of the microchannel, and the penetration rate (or rate of hyphal extension) was calculated for each hypha or group of hyphae as it/they moved through the channel at each time point as the change in penetration length from the previous time point, divided by the elapsed time between images. Images were recorded from the time that hyphae entered individual channels to the time in which hyphae exited the channels or stopped growing (i.e., exhibited no change in penetration over three-time intervals). The penetration rates measured in the linear microchannels for different width channels were plotted for each species at each time interval (Fig 4). Overall, the penetration rates of all the species tested tended to decrease as they moved through the microchannel geometry. *M. sextalata*, *P. minutissima*, *L. elongate*, *Beniella*, *S.* sp., and *L. bicolor* decreased as they moved through the microchannels geometry, for both the 5 and 10 μm wide channels. *F. falciforme* showed marginal changes in penetration rates, once entering confined channels for both channel widths.

## Comparison of penetration rates in 5 and 10—μm wide microchannels

The measured penetration rates of 5 of the 7 species for both the 5 and 10 μm channels are shown in the histogram plots in Fig 2. *L. bicolor* and *S.* sp. are not shown because all rates were measured to be less than 1 μm/min. The time for each strain to reach the entrance of the microchannels varied based on their intrinsic growth rate, size of the colony, and orientation of fungal agar which is reported in Table 1.

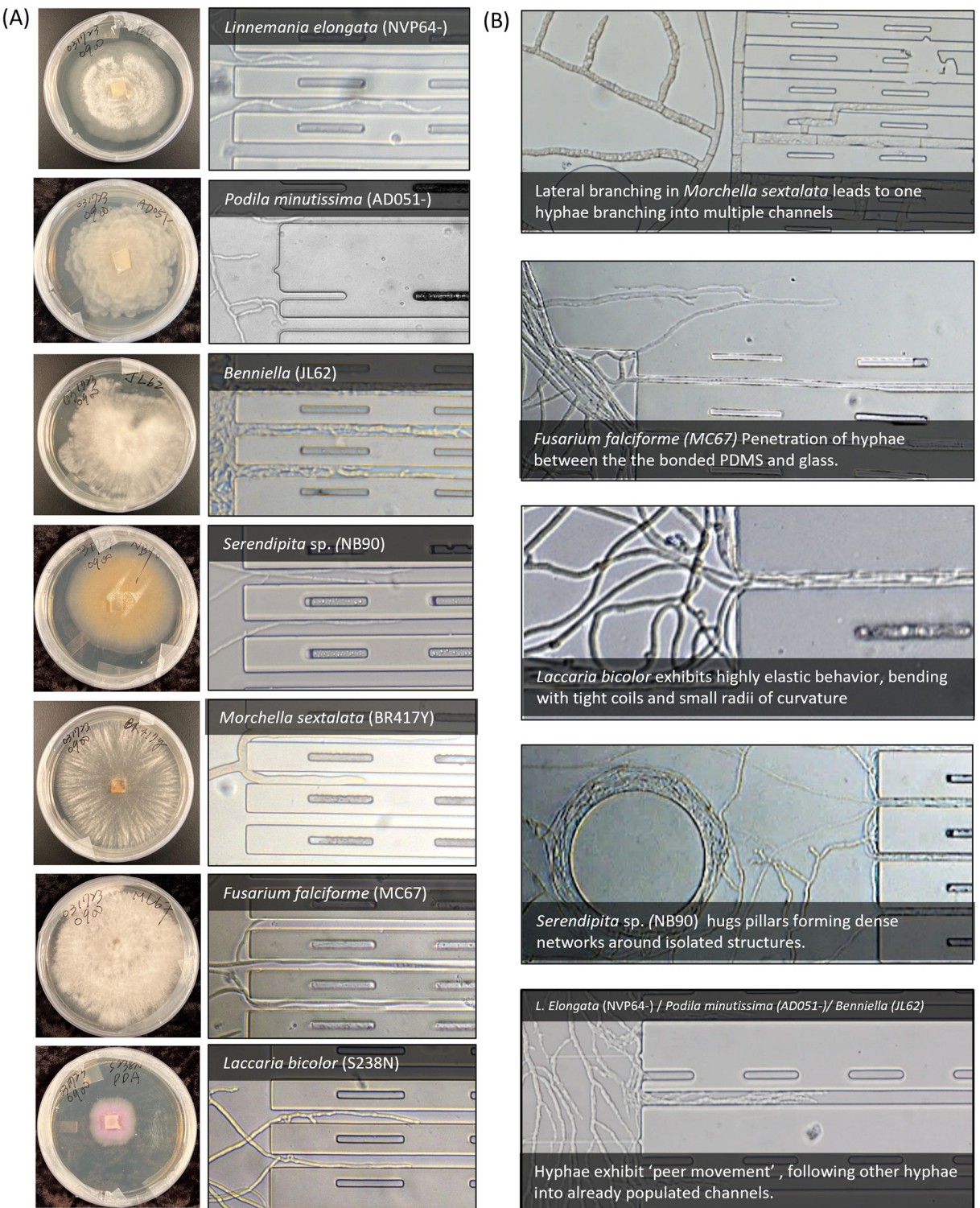

**Fig 3.** Characterization of fungi in engineered microhabitats (A) Fungal isolates inoculated on PDA plates. Mycelial network of fungal isolates entering microhabitat channels from open geometries (B) Representative images of mycelial characteristics from selected species.

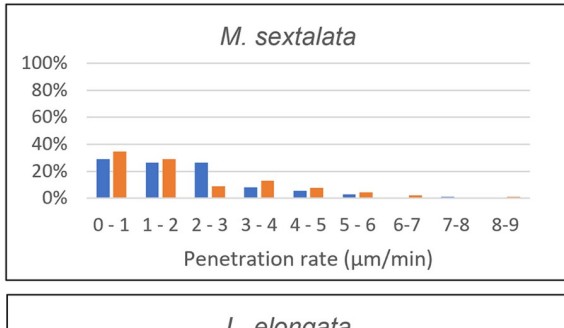

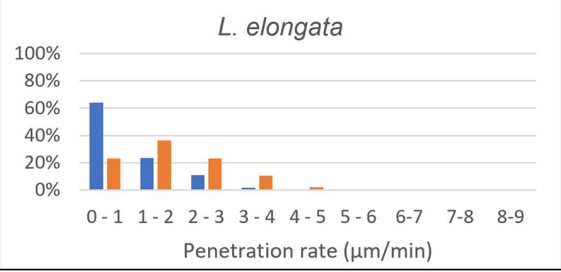

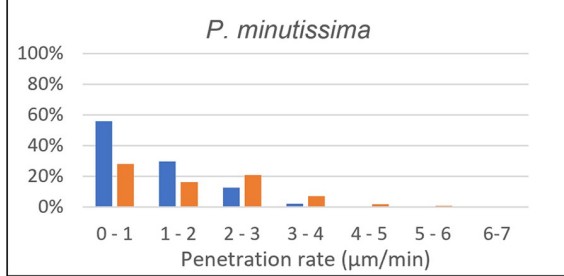

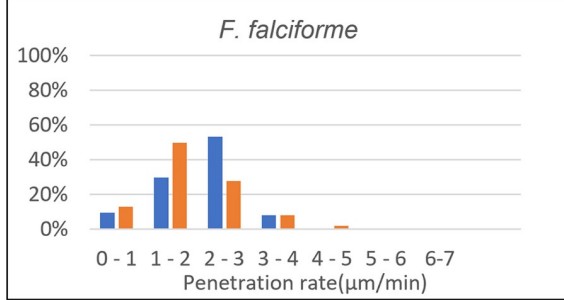

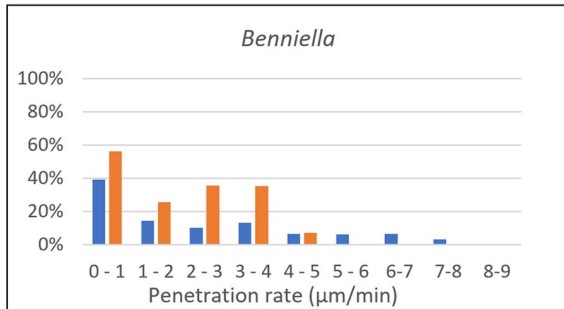

**Fig 4. Histogram plots of instantaneous growth rates.** Changes in instantaneous growth rates are evident and illustrate the impact of increasing confinement on fungal hyphal penetration rates.

The distribution of 'instantaneous' penetration rates was compared between 5- and 10-μm wide microchannels. The penetration rates of all species tested, with the exception of *F. falciforme* were higher when moving through the 10-μm wide microchannels and showed more instances of higher penetration rates compared to the 5-μm wide microchannels. A student's two-tailed T-test for unequal variances indicates significant differences between 5- and 10-μm wide channels, for all species except *M. sextalata* (Table 1), reinforcing the impact of confinement on hyphal growth. In contrast, *F. falciforme* demonstrated instances of higher rates of penetration while moving through 10-μm wide microchannels than the 5-μm wide channels. To reiterate, the measured penetration rates of each strain under confinement in 5- versus 10-μm channels all show statistically significant differences (p-value < 0.005), with the exception of *M. sextalata* (p-value = 0.01) supporting the hypothesis that confinement impacts hyphal penetration rates.

The "macroscale" expansion rates for each fungal species were measured and are also reported in Table 1. All the strains were inoculated at the center of Petri dishes individually over time until their mycelium reached the edge of the dish or grew for up to 14 days. The expansion rates at the macro-scale were considerably higher than the values reported for penetration through microchannels [18].

## Discussion

Soil fungi of different phyla share common features when cultured in microfluidics, which can be seen as potential abilities in the real plant-soil interface (see Table 1). For example, the Ascomycota isolates tested, *F. falciforme* and *M. sextalata* had a strong enough tip force strength to break the bonding of the PDMS devices. *F. falciforme* produces copious amounts of mitotic spores and shows its ability to break the bonding of microfluidics as a form of fungal spores (observed, data not included). Representing Mucoromycota, *L. elongate*, *P. minutissima*, and *Benniella* have branch frequently and exhibit peer movements, in which groups of hyphae move through similar channels in unison under confinement. *L. bicolor* and *S.* sp. represent the Basidiomycota. *L. bicolor* was the slowest grower in this study, but also demonstrates its great elasticity against physical obstacles and turns into a coil-formation morphology to look for more space and nutrients when navigating inside the space. *S.* sp. was another slower grower and appeared to reduce growth due to the spore formation at their hyphal tips. This isolate also demonstrated a unique pillar-hugging behavior while moving around the obstacles.

Generally, soil fungi from those species examined in this work exhibit similar characteristics when incubated in microfluidic devices. Certainly, across a given mycelium network broad ranges in hyphal penetration rates were observed, with considerable changes in which hypha or regions of the mycelial network were growing at a given time. For example, in some experiments, hyphae penetrated and exited the capillary channels quickly while other regions of the mycelial network halted growth. This highlights the potential to use these microhabitats to explore how fungi allocate resources to respond to physical and chemical changes in their environment and enhance their ability to explore complex spaces for nutrients This supports work suggesting that within the mycelium network, individual hyphae may have heterogeneous roles in decision-making as a general unit of mycelium [20]. Fungal hyphal growth was consistently impacted by the growth of other local hyphal, tracking the path of other hyphae when navigating the local environments. Whether this is the result of local chemical signaling, nutrient depletion, or mechanical influence remains a significant question. Future work to determine the nature of these interactions is warranted. Additionally, after entering the capillary channels, hyphae started to exhibit more branching, forming additional hyphae in the

confined space. The penetration and branching of hyphae pushing inside confined channels may mimic an environment where the mycelium is likely to physically expand its territory by pushing hyphal tips when surrounded by soil aggregates and plant roots in micro-pore regions. Optical interference from the microchannel side walls made quantitative analysis of these observations difficult, however understanding the impact of confinement and mechanical cell wall stress on the induction of branching is an opportunity for exploration.

## Conclusions

Microfluidic habitats were used to serve as a platform for examining the impact of confinement on the penetration of the mycelium of soil fungi. Species of Ascomycota, Mucoromycota, and Basidiomycota slowed their penetration rates after entering the microchannels, with increased confinement in smaller 5-µm channels showing slower penetration than in 10-µm channels. The exceptions to this were *F. falciforme*, which increased its average penetration rate in smaller channels, and *L. bicolor*, which exhibited no change in penetration rates between the two channel dimensions that were tested. All the species showed dramatic differences between their microchannel penetration rates and 'macroscale' expansion in Petri dishes. Unique behaviors for these species were observed and captured in optical micrographs and highlighted the potential for further studies that utilize unique channel geometries and spatiotemporal chemical gradients to examine resource allocation, chemotaxis, and response to different physical and chemical cues.

The well-defined microchannel dimensions and layout allow careful control of nutrients and water to fungal hyphae. In situ, metabolite profiling may help uncover the decision-making mechanism of fungal mycelium. Additionally, with the addition of bacteria and plants into platforms described here, cross-kingdom signaling and physical interactions that allow direct observation of rhizosphere processes may be realized.

## Acknowledgments

We are grateful to Jessy L. Labbé for giving fungal inoculation training. We thank Kevin Lester for fabricating 3D-printed sample inserts.

## Author Contributions

**Conceptualization:** Yi-Syuan Guo, Julian A. Liber, Jennifer L. Morrell-Falvey, Gregory Bonito, Scott T. Retterer.

**Data curation:** Yi-Syuan Guo, Julian A. Liber.

**Formal analysis:** Yi-Syuan Guo.

**Funding acquisition:** Jennifer L. Morrell-Falvey, Scott T. Retterer.

**Investigation:** Yi-Syuan Guo, Julian A. Liber.

**Methodology:** Yi-Syuan Guo, Jennifer L. Morrell-Falvey, Gregory Bonito, Scott T. Retterer.

**Project administration:** Jennifer L. Morrell-Falvey.

**Resources:** Jennifer L. Morrell-Falvey, Gregory Bonito.

**Supervision:** Jennifer L. Morrell-Falvey, Gregory Bonito, Scott T. Retterer.

**Validation:** Yi-Syuan Guo.

**Visualization:** Yi-Syuan Guo.

**Writing – original draft:** Yi-Syuan Guo, Scott T. Retterer.

**Writing – review & editing:** Yi-Syuan Guo, Jennifer L. Morrell-Falvey, Gregory Bonito.

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
