## [Decision Letter · Decision Letter 0]

17 Sep 2024

PONE-D-24-24805Monitoring the impact of confinement on hyphal penetration and fungal behaviorPLOS ONE

Dear Dr. Retterer,

Thank you for submitting your manuscript to PLOS ONE. After careful consideration, we feel that it has merit but does not fully meet PLOS ONE’s publication criteria as it currently stands. Therefore, we invite you to submit a revised version of the manuscript that addresses the points raised during the review process.

The reviewer raised the point of missing statistics. If those can be provided and the other points can be cleared, the publication may be considered.

We look forward to receiving your revised manuscript.

Kind regards,

Erika Kothe

Academic Editor

PLOS ONE

Journal Requirements: When submitting your revision, we need you to address these additional requirements. 1. Please ensure that your manuscript meets PLOS ONE's style requirements, including those for file naming. The PLOS ONE style templates can be found at https://journals.plos.org/plosone/s/file?id=wjVg/PLOSOne_formatting_sample_main_body.pdf and https://journals.plos.org/plosone/s/file?id=ba62/PLOSOne_formatting_sample_title_authors_affiliations.pdf 2. Thank you for stating the following financial disclosure: "This work was supported by the Office of Science, Biological and Environmental Research, as part of the Plant Microbe Interfaces Scientific Focus Area (https://pmi.ornl.gov), Department of Energy, BER, ERKP730.   Fabrication of Microfluidic Devices was performed at the Center for Nanophase Materials Sciences, A DOE Nanoscience User Facility, Department of Energy, BES ERKCZO1" Please state what role the funders took in the study.  If the funders had no role, please state: ""The funders had no role in study design, data collection and analysis, decision to publish, or preparation of the manuscript."" If this statement is not correct you must amend it as needed. Please include this amended Role of Funder statement in your cover letter; we will change the online submission form on your behalf. 3. We note that your Data Availability Statement is currently as follows: All relevant data are within the manuscript and its Supporting Information files. Please confirm at this time whether or not your submission contains all raw data required to replicate the results of your study. Authors must share the “minimal data set” for their submission. PLOS defines the minimal data set to consist of the data required to replicate all study findings reported in the article, as well as related metadata and methods (https://journals.plos.org/plosone/s/data-availability#loc-minimal-data-set-definition). For example, authors should submit the following data: - The values behind the means, standard deviations and other measures reported;- The values used to build graphs;- The points extracted from images for analysis. Authors do not need to submit their entire data set if only a portion of the data was used in the reported study. If your submission does not contain these data, please either upload them as Supporting Information files or deposit them to a stable, public repository and provide us with the relevant URLs, DOIs, or accession numbers. For a list of recommended repositories, please see https://journals.plos.org/plosone/s/recommended-repositories. If there are ethical or legal restrictions on sharing a de-identified data set, please explain them in detail (e.g., data contain potentially sensitive information, data are owned by a third-party organization, etc.) and who has imposed them (e.g., an ethics committee). Please also provide contact information for a data access committee, ethics committee, or other institutional body to which data requests may be sent. If data are owned by a third party, please indicate how others may request data access.

Reviewers' comments:

Reviewer's Responses to Questions

**Comments to the Author**

1. Is the manuscript technically sound, and do the data support the conclusions?

Reviewer #1: No

2. Has the statistical analysis been performed appropriately and rigorously? 

Reviewer #1: Yes

3. Have the authors made all data underlying the findings in their manuscript fully available?

Reviewer #1: Yes

4. Is the manuscript presented in an intelligible fashion and written in standard English?

Reviewer #1: Yes

5. Review Comments to the Author

Reviewer #1: The reviewed studies are undoubtedly of great theoretical importance. They significantly expand knowledge about the biology and behavior of fungi and the relationships between fungi and plants. They may also have great practical importance, for example in agriculture.

Line 262-264: The Authors write …’ The penetration rates measured in the linear microchannels for different width channels were plotted for each species at each time interval (Figure 4). Overall, the penetration rates of all the species tested tended to decrease as they moved through the microchannel geometry.’ Were the obtained differences statistically significant?

6. PLOS authors have the option to publish the peer review history of their article (what does this mean?). If published, this will include your full peer review and any attached files.

Reviewer #1: No

---

## [Author Response · Author response to Decision Letter 0]

11 Oct 2024

The reviewer acknowledged the significance of the work noting the “great theoretical importance” of the work as well as the “great practical importance” of the study. However, the reviewer questioned whether the results shown were “statistically significant”. In the previous version of the manuscript we had, in fact, noted that a student’s two tailed t-test was used to gauge the statistical significance of the difference between the penetration rates measured in the 5-µm versus the 10-µm wide microchannels. With p-values well below 0.005, as reported in the manuscript, the t-test supports that the observed differences were statistically significant. While all this information was reported in the original submission, we acknowledge that the text could have been clearer, and that additional information about the number of experimental replicates performed and number of measurements used to calculate these p-values would have been beneficial to the reviewer. Changes to the text and appropriate tables were made to improve the clarity of the portions of the article that described those aspects of the study and the subsequent analysis of the results. We thank the reviewer for their efforts and feel that we have addressed their concerns.

---

## [Editor Report · Decision Letter 1]

15 Oct 2024

Monitoring the impact of confinement on hyphal penetration and fungal behavior

PONE-D-24-24805R1

Dear Dr. Retterer,

We’re pleased to inform you that your manuscript has been judged scientifically suitable for publication and will be formally accepted for publication once it meets all outstanding technical requirements.

Kind regards,

Erika Kothe

Academic Editor

PLOS ONE
---

## [Editor Report · Acceptance letter]

19 Oct 2024

PONE-D-24-24805R1 

PLOS ONE

Dear Dr. Retterer, 

I'm pleased to inform you that your manuscript has been deemed suitable for publication in PLOS ONE. Congratulations! Your manuscript is now being handed over to our production team.

Kind regards, 

on behalf of

Prof. Dr. Erika Kothe 

Academic Editor

PLOS ONE
